# GILA: GRAPH STRUCTURE LEARNING FOR CLASS-IMBALANCED TABULAR DATA

## ABSTRACT

Graph-based learning has demonstrated strong performance across domains by capturing inter-sample dependencies, often surpassing traditional methods focused only on feature-level patterns. However, in tabular data, the relationships between instances are typically implicit or undefined, making it difficult to directly apply graph-based methods. Graph Structure Learning (GSL) addresses this by constructing task-relevant graph structures that explicitly capture the relationships from the data. Nonetheless, a critical yet underexplored challenge in tabular GSL is class imbalance, which is commonly encountered in real-world applications. In this paper, we propose **G**raph Structure Learning for Class-**I**mbalanced Dat**a** (**GILA**) to overcome the difficulties of small-scale and imbalanced tabular datasets. GILA introduces *helper nodes* to assist the minority class. These helper nodes are optimized using an *updater*, and promote class separation among their connected neighbors by influencing their representations through message propagation. Our experiments on 44 imbalanced tabular datasets demonstrate that GILA outperforms existing tabular GSL models. Even under severe class overlap, it still achieves strong performance by leveraging helper nodes that facilitate class discrimination. Furthermore, visualization of the embedding space highlights how helper nodes guide the separation of minority-class samples from majority ones. Code for this submission is provided in an anonymous repository at: https://github.com/anon-gila/GILA

## 1 INTRODUCTION

Graph data are widely used in various domains, such as social networks Tang & Liu (2010); Tian et al. (2025), transportation systems James (2022), and recommendation systems Huang et al. (2021); Wu et al. (2022). Each instance is represented as a node, and the explicit relationships between instances are expressed through edges. These data have demonstrated significant performance gains with Graph Neural Networks (GNNs) Kipf & Welling (2016); Velickovic et al. (2017); Xu et al. (2018), which leverage inter-sample interactions, in contrast to conventional methods that treat each sample independently and focus solely on feature interactions Wu et al. (2023); Liu et al. (2022). However, in the case of tabular data, the relationships between samples are not explicitly defined, making it difficult to directly apply graph-based methods.

Recently, several studies have reported improved performance by constructing graph structures based on latent relationships learned among samples in tabular data Guo et al. (2021); Liao & Li (2023). Unlike traditional tabular models that focus solely on feature-level patterns, these approaches aim to capture richer information by modeling implicit inter-instance relationships, as illustrated in Figure 1. However, existing tabular Graph Structure Learning (GSL) models tend to overlook a critical challenge in real-world tabular data applications: events of interest are often rare. For example, in disease prediction tasks, the incidence rate is typically low, and in defect detection

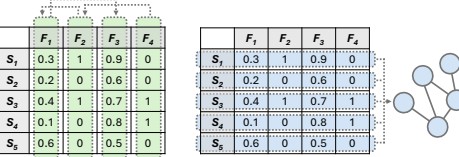

(a) Feature interaction    (b) Instance interaction

Figure 1: Traditional tabular models rely solely on feature-level interactions (a), whereas graph-based models leverage both feature- and instance-level leveraging inter-instance relationships for richer context (b).

scenarios, defective items may constitute less than 1% of all samples. Nevertheless, most existing GSL models assume balanced class distributions. Applying such models directly to imbalanced data can result in both graph structures and classifiers being biased toward the majority class, ultimately leading to poor performance on minority class prediction.

In this work, we propose a Graph Structure Learning (GSL) framework to improve classification performance on imbalanced tabular data by learning latent inter-sample relationships and introducing *helper nodes* to mitigate class imbalance. Our method, called **G**raph Structure Learning for Class-**I**mbalanced Dat**a** (GILA), specifically targets two challenging scenarios: small-scale datasets and class imbalance. Deep learning methods usually require large datasets to accurately learn the underlying data distribution. However, in many real-world tabular applications, data is often too scarce to support reliable distribution estimation. Instead of relying on distribution estimation, we leverage GSL to extract richer inter-sample information, enabling more effective learning from small-scale tabular data. To further combat class imbalance, GILA introduces *helper nodes* for the minority class, which are optimized using an *updater* module. These helper nodes enhance the separability of the minority class from the majority class by influencing the representations of their connected neighbors through message propagation. Our main contributions are summarized as follows:

- To the best of our knowledge, we are the first to propose graph structure learning to deal with the class imbalance problem in small-scale tabular data. We introduce *helper nodes* for minority classes and optimize them using an *updater*.

- We validate the effectiveness of our method through experiments on 44 real-world imbalanced tabular datasets. Furthermore, a correlation analysis between class overlap and GSL model performance demonstrates that GILA remains robust even in highly overlapping feature spaces, where distinguishing between classes is inherently difficult.

- We provide visual evidence to illustrate how the helper nodes influence the representations of their neighbors through message propagation, thereby promoting class separation in the embedding space.

## 2 RELATED WORK AND PRELIMINARY

### 2.1 GRAPH STRUCTURE LEARNING

GNNs are highly sensitive to the graph structure, and the presence of spurious or noisy connections can significantly degrade prediction performance Dai et al. (2018). Graph Structure Learning (GSL) has emerged as a solution to refine—or even reconstruct—the graph structure, rather than directly use the one initially provided Jin et al. (2020); Chen et al. (2020); Fatemi et al. (2021). Furthermore, there has been increasing research interest in inferring graph structures from non-graph data, such as tabular datasets Liao & Li (2023); Zhou et al. (2022).

According to Zhu et al. (2021), many existing GSL strategies consist of three steps: (1) initial graph construction, (2) structure modeling, and (3) representation learning and prediction.

**Initial Graph Construction** If the original graph is unavailable, some GSL methods adopt $k$ Nearest Neighbors ($k$-NN) graphs Preparata & Shamos (2012) or $\epsilon$ proximity thresholding ($\epsilon$-graphs) Bentley et al. (1977) as the initial graph. Both are based on pairwise distances. The $k$-NN graph method connects the $k$-nearest node pairs, while the $\epsilon$-graph method generates edges based on a given threshold.

**Structure Modeling** In this step, the Structure Learner refines or infers a task-optimized adjacency matrix. Structure modeling can be broadly categorized into three approaches: *Metric-based approach*, *Neural approach*, and *Direct approach*. In the *Metric-based approach*, similarity is calculated using a metric function to determine edge weights. A graph structure satisfying the homophily assumption is created, where similar nodes have higher probabilities of being connected. The *Neural approach* utilizes neural networks to learn edge weights based on node features, enabling the capture of complex and non-linear patterns. The *Direct approach* defines the adjacency matrix as a set of learnable parameters, offering greater flexibility as it does not rely on node representations.

**Representation Learning and Prediction** In the representation learning and prediction phase, GNNs are trained using the learned adjacency matrix to obtain the node representations $\mathbf{Z}$ for the

prediction task. $\mathbf{Z}$ may also be reused in the structure modeling step in some methods. The formulation of the GNN model used for prediction is provided in the following section.

## 2.2 GRAPH NEURAL NETWORKS

Graph Neural Networks (GNNs) Kipf & Welling (2016); Velickovic et al. (2017); Xu et al. (2018) have emerged as powerful approaches that capture relational information of graph data and node features comprehensively. GNNs utilize a message passing mechanism to iteratively aggregate neighbor information and propagate it. A graph can be represented as $\mathcal{G} = (\mathcal{V}, \mathbf{A}, \mathbf{X})$, where $\mathcal{V} = \{v_1, v_2, \ldots, v_n\}$ denotes the set of nodes, $n$ is the number of nodes, and $\mathbf{A} \in \mathbb{R}^{n \times n}$ is the adjacency matrix. If $v_i$ and $v_j$ are connected, the component $a_{ij}$ of the adjacency matrix is assigned 1; otherwise 0. $\mathbf{X} \in \mathbb{R}^{n \times d}$ is the set of $x_i$, which is the $d$-dimensional feature vector of node $v_i$. Graph Convolutional Network (GCN) Kipf & Welling (2016), which is a powerful variant of GNNs, can be formalized as follows:

$$\mathbf{H}^{(l)} = \sigma(\tilde{\mathbf{D}}^{-\frac{1}{2}} \hat{\mathbf{A}} \tilde{\mathbf{D}}^{-\frac{1}{2}} \mathbf{H}^{(l-1)} \mathbf{W}^{(l)}) \tag{1}$$

where $\hat{\mathbf{A}} = \mathbf{A} + \mathbf{I}$ corresponds to an adjacency matrix with self-loops, $\mathbf{I}$ is the identity matrix, $\tilde{\mathbf{D}}$ is the degree matrix of $\hat{\mathbf{A}}$, $\mathbf{H}^{(l)}$ represents the node embeddings at the $l$-th layer, $\mathbf{W}^{(l)}$ is the weight matrix at the $l$-th layer, and $\sigma(\cdot)$ denotes the activation function like ReLU.

## 2.3 IDGL

In this study, we employ IDGL Chen et al. (2020) to learn the graph structure for tabular data. As described in Section 2.1, IDGL simultaneously carries out Structure Modeling and Representation Learning and Prediction in a joint and iterative manner. The process of IDGL can be summarized as follows:

IDGL utilizes a multi-head version similarity metric to learn the adjacency matrix.

$$s_{ij}^h = \cos(\mathbf{w}_h \odot \mathbf{v}_i, \mathbf{w}_h \odot \mathbf{v}_j), \quad s_{ij} = \frac{1}{N} \sum_{h=1}^{N} s_{ij}^h \tag{2}$$

where $s_{ij}^h$ is the $h$-th head weighted cosine similarity, $\odot$ denotes the Hadamard product, and $\mathbf{w}$ is a learnable weight vector with the same dimensionality as the node feature vector. $\mathbf{v}$ represents the input feature or node embedding. $N$ weight vectors are used to calculate $N$ number of similarity matrices, and their average is taken as the final similarity. A symmetric, non-negative adjacency matrix $\mathbf{A}$ is constructed from the similarity matrix $\mathbf{S} = [s_{ij}]_{i,j=1}^n \in \mathbb{R}^{n \times n}$ via $\varepsilon$-neighborhood thresholding ($\varepsilon \geq 0$) and symmetrization. The final adjacency matrix $\tilde{\mathbf{A}}$ is computed by combining the learned graph with the initial graph.

$$\tilde{\mathbf{A}}^{(t)} = \lambda \mathbf{L}^{(0)} + (1 - \lambda)\{\eta f(\mathbf{A}^{(t)}) + (1 - \eta) f(\mathbf{A}^{(1)})\} \tag{3}$$

where $t$ denotes the iteration step, $\mathbf{L}^{(0)} = \mathbf{D}^{(0)^{-1/2}} \mathbf{A}^{(0)} \mathbf{D}^{(0)^{-1/2}}$ represents the normalized adjacency matrix of the initial graph $\mathbf{A}^{(0)}$. $\mathbf{D}^{(0)}$ denotes the degree matrix of $\mathbf{A}^{(0)}$. $\mathbf{A}^{(1)}$ is constructed from the initial node features $\mathbf{X}$, whereas $\mathbf{A}^{(t)}$ is derived from the previously updated node embeddings $\mathbf{Z}^{(t-1)}$ learned by the GNN. When there is no initial graph available like tabular data, IDGL generates a $k$-NN-based initial graph. The overall loss is denoted $\mathcal{L} = \mathcal{L}_{\text{pred}} + \mathcal{L}_{\mathcal{G}}$, where $\mathcal{L}_{\text{pred}}$ represents the prediction loss and $\mathcal{L}_{\mathcal{G}}$ is the graph regularization loss. The graph regularization loss aims to control the smoothness, connectivity, and sparsity of the learned graph.

## 3 METHOD

Consider an imbalanced tabular dataset depicted as feature matrix $\mathbf{X} \in \mathbb{R}^{n \times d}$ with labels $\mathbf{y} \in \mathbb{R}^n$, where $n$ represents the number of data samples and $d$ denotes the feature dimension. The goal of this study is to improve the classification performance of the minority class in situations with small-scale and highly imbalanced datasets. To address this, we convert the tabular data $\mathbf{X}$ into a task-optimal graph $\mathcal{G}^* = (\mathcal{V}, \mathbf{A}^*, \mathbf{X}^*)$ using GSL. We also introduce pseudo-nodes, referred to as *helper nodes*, to assist the minority class. The helper nodes are optimized to make the minority class representations more distinguishable from those of the majority class. The overall framework of the proposed method is illustrated in Figure 2.

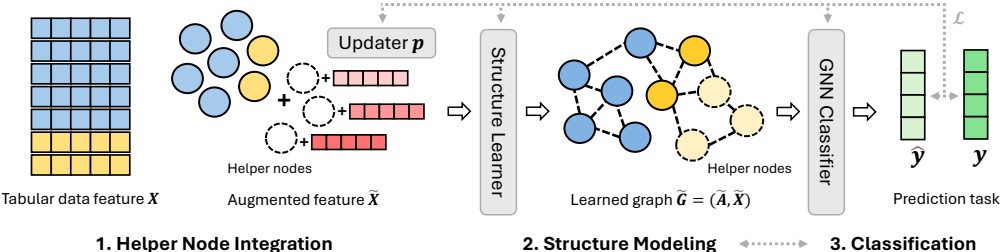

Figure 2: Overview of the proposed method GILA. The overall loss $\mathcal{L}$ is used to jointly optimize all parameters — the updater $\mathbf{p}$, structure learner, and GNN classifier — at each epoch.

## 3.1 HELPER NODE INTEGRATION

**Introducing Helper Node** We introduce helper nodes, which are pseudo-nodes designed to support the minority class. Each feature vector of the helper node $\mathbf{x}^h$ is initialized with a zero vector, and the feature matrix is extended to $\mathbf{X}' \in \mathbb{R}^{(n+m) \times d}$, where $m$ represents the number of helper nodes.

$$
\begin{aligned}
\mathbf{X} &= \{\mathbf{x}_1, \ldots, \mathbf{x}_n\} \\
\mathbf{X}' &= \{\mathbf{x}_1, \ldots, \mathbf{x}_n, \mathbf{x}_1^h, \ldots, \mathbf{x}_m^h\}
\end{aligned}
\tag{4}
$$

Helper nodes are assigned the label of the minority class. Note that the helper nodes are included only in the training set, not in the validation or test sets. In the training set, the number of minority class samples, including the helper nodes, is equal to the number of majority class samples. For example, if the number of nodes in the minority class is 10 fewer than that of the majority class, we add 10 helper nodes and assign them to the minority class to balance the class distribution.

**Update Helper Nodes** The helper nodes are updated by the learnable component. Specifically, each helper node has its own *updater* $\mathbf{p}$, with the same dimension $d$ as the node feature.

$$
\mathbf{p}_1, \mathbf{p}_2, \ldots, \mathbf{p}_m \in \mathbb{R}^d
\tag{5}
$$

$$
\begin{aligned}
\mathbf{X}' &= \{\mathbf{x}_1, \ldots, \mathbf{x}_n, \mathbf{x}_1^h, \ldots, \mathbf{x}_m^h\} \\
\tilde{\mathbf{X}} &= \{\mathbf{x}_1, \ldots, \mathbf{x}_n, \mathbf{x}_1^h + \mathbf{p}_1, \ldots, \mathbf{x}_m^h + \mathbf{p}_m\}
\end{aligned}
\tag{6}
$$

The learnable vector $\mathbf{p}$ is element-wise added to each helper node in the feature space. The augmented feature matrix $\tilde{\mathbf{X}} \in \mathbb{R}^{(n+m) \times d}$, which incorporates both original nodes and helper nodes, is used for the structure modeling. The updater $\mathbf{p}$ are jointly optimized with the structure learner and the GNN classifier (see Section 3.2 and 3.3). The goal of training the updater $\mathbf{p}$ is to optimize the helper nodes so that the representations of the nodes connected to them become distinguishable from those of other classes.

## 3.2 STRUCTURE MODELING BY STRUCTURE LEARNER

In this step, we use IDGL as the structure learner for structure modeling. IDGL adopts a co-training strategy that jointly updates the structure learner and the GNN classifier, making it well-suited for simultaneously optimizing the helper nodes and the overall graph structure.

$$
\tilde{\mathbf{A}}^{(t)} = f_{\theta_{\mathrm{GSL}}}(\tilde{\mathbf{X}})
\tag{7}
$$

As shown in equation 3, $\tilde{\mathbf{A}}^{(t)}$ is computed from $\mathbf{A}^{(0)}$, $\mathbf{A}^{(1)}$, and $\mathbf{A}^{(t)}$. Note that IDGL constructs $\mathbf{A}^{(1)}$ from the initial node features $\mathbf{X}$, whereas our framework employs the augmented feature matrix $\tilde{\mathbf{X}}$, which incorporates helper nodes. $\mathbf{A}^{(t)}$ is derived from the previously updated node embeddings $\mathbf{Z}^{(t-1)}$ learned by the GNN classifier, which will be detailed in Section 3.3.

Table 1: Information of datasets

| Idx | Datasets | # Samples | # Attrs | IR | $R_{\mathrm{aug}}$ | Idx | Datasets | # Samples | # Attrs | IR | $R_{\mathrm{aug}}$ |
|---|---|---|---|---|---|---|---|---|---|---|---|
| 1 | Ecoli034vs5 | 200 | 7 | 9.00 | 0.225 | 23 | Glass0146vs2 | 205 | 9 | 11.06 | 0.918 |
| 2 | Yeast2vs4 | 514 | 8 | 9.08 | 0.336 | 24 | Glass2 | 214 | 9 | 11.59 | 0.921 |
| 3 | Ecoli067vs35 | 222 | 7 | 9.09 | 0.328 | 25 | Ecoli0147vs56 | 332 | 6 | 12.28 | 0.222 |
| 4 | Ecoli0234vs5 | 202 | 7 | 9.10 | 0.225 | 26 | Cleveland0vs4 | 177 | 13 | 12.62 | 0.926 |
| 5 | Glass015vs2 | 172 | 9 | 9.12 | 0.903 | 27 | Ecoli0146vs5 | 280 | 6 | 13.00 | 0.232 |
| 6 | Yeast0359vs78 | 506 | 8 | 9.12 | 0.650 | 28 | Shuttlec0vsc4 | 1,829 | 9 | 13.87 | 0.008 |
| 7 | Yeast0256vs3789 | 1,004 | 8 | 9.14 | 0.411 | 29 | Yeast1vs7 | 459 | 7 | 14.30 | 0.903 |
| 8 | Yeast02579vs368 | 1,004 | 8 | 9.14 | 0.183 | 30 | Glass4 | 214 | 9 | 15.47 | 0.435 |
| 9 | Ecoli046vs5 | 203 | 6 | 9.15 | 0.225 | 31 | Ecoli4 | 336 | 7 | 15.80 | 0.188 |
| 10 | Ecoli01vs235 | 244 | 7 | 9.17 | 0.338 | 32 | Page_blocks13vs4 | 472 | 10 | 15.86 | 0.371 |
| 11 | Ecoli0267vs35 | 224 | 7 | 9.18 | 0.328 | 33 | Abalone918 | 731 | 8 | 16.40 | 0.898 |
| 12 | Glass04vs5 | 92 | 9 | 9.22 | 0.702 | 34 | Glass016vs5 | 184 | 9 | 19.44 | 0.952 |
| 13 | Ecoli0346vs5 | 205 | 7 | 9.25 | 0.226 | 35 | Shuttlec2vsc4 | 129 | 9 | 20.50 | 0.795 |
| 14 | Ecoli0347vs56 | 257 | 7 | 9.28 | 0.217 | 36 | Yeast1458vs7 | 693 | 8 | 22.10 | 0.957 |
| 15 | Yeast05679vs4 | 528 | 8 | 9.35 | 0.657 | 37 | Glass5 | 214 | 9 | 22.78 | 0.959 |
| 16 | Vowel0 | 988 | 13 | 9.98 | 0.030 | 38 | Yeast2vs8 | 482 | 8 | 23.10 | 0.431 |
| 17 | Ecoli067vs5 | 220 | 6 | 10.00 | 0.273 | 39 | Yeast4 | 1,484 | 8 | 28.10 | 0.833 |
| 18 | Glass016vs2 | 192 | 9 | 10.29 | 0.912 | 40 | Yeast1289vs7 | 947 | 8 | 30.57 | 0.968 |
| 19 | Ecoli0147vs2356 | 336 | 7 | 10.59 | 0.315 | 41 | Yeast5 | 1,484 | 8 | 32.73 | 0.309 |
| 20 | Leddigit02456789vs1 | 443 | 7 | 10.97 | 0.669 | 42 | Ecoli0137vs26 | 281 | 7 | 39.14 | 0.279 |
| 21 | Ecoli01vs5 | 240 | 6 | 11.00 | 0.229 | 43 | Yeast6 | 1,484 | 8 | 41.40 | 0.474 |
| 22 | Glass06vs5 | 108 | 9 | 11.00 | 0.918 | 44 | Abalone19 | 4,174 | 8 | 129.44 | 0.992 |

## 3.3 CLASSIFICATION BY GNN CLASSIFIER

The augmented feature matrix $\tilde{\mathbf{X}}$ and the learned adjacency matrix $\tilde{\mathbf{A}}$ are used to train the GNN classifier. While various GNN models can be employed for node classification, we adopt the GCN in our experiments.

$$
\mathbf{Z}^{(t)} = \mathbf{ReLU}(f_{\theta_{\mathrm{GNN}_1}}(\tilde{\mathbf{X}}, \tilde{\mathbf{A}}^{(t)}))
$$
$$
\hat{\mathbf{y}} = \mathbf{softmax}(f_{\theta_{\mathrm{GNN}_2}}(\mathbf{Z}^{(t)}, \tilde{\mathbf{A}}^{(t)}))
$$
(8)

The overall loss of IDGL is defined as the sum of the prediction loss $\mathcal{L}_{\mathrm{pred}}$ and the graph regularization loss $\mathcal{L}_{\mathcal{G}}$, i.e., $\mathcal{L} = \mathcal{L}_{\mathrm{pred}} + \mathcal{L}_{\mathcal{G}}$. Here, $\mathcal{L}_{\mathcal{G}}$ corresponds to the loss used in the structure modeling step, while $\mathcal{L}_{\mathrm{pred}}$ is used for the classification task. In each training epoch, the overall loss $\mathcal{L}$ is backpropagated through the entire framework, so that the updater parameters $\{\mathbf{p}_i\}_{i=1}^m$, the structure learner parameters $\theta_{\mathrm{GSL}}$, and the GNN classifier parameters $\theta_{\mathrm{GNN}}$ are jointly updated.

In summary, the structure learner models the relations among all nodes, including helper nodes. The prediction results then influence the helper nodes, which in turn guide the structure learner to capture a more appropriate graph structure. Moreover, through the message-passing function of the GNN classifier, helper nodes affect the representations of their neighbors. To achieve more accurate classification, the helper nodes are trained such that the nodes connected to them become distinguishable from those of other classes. This entire process is carried out iteratively, with the components continuously refining one another.

## 4 EXPERIMENT

### 4.1 EXPERIMENTAL SETTINGS

**Datasets** The 44 small-scale and imbalanced datasets from the KEEL repository[1] Derrac et al. (2015) are used for our experiment, with detailed information in Table 1. The Imbalance Ratio (IR) is defined as $\mathrm{IR} = N_{\mathrm{Maj}}/N_{\mathrm{Min}}$, where $N_{\mathrm{Maj}}$ and $N_{\mathrm{Min}}$ denote the number of samples in the majority and minority classes, respectively. An IR of 1 implies that the two classes are balanced, while an IR of 9 indicates that the majority class contains 9 times more samples than the minority class. The 44 datasets are severely imbalanced, with an IR ranging from 9 to 129.44. The augmented R-value ($R_{\mathrm{aug}}$) Fu et al. (2020) quantifies the degree of class overlap in the original feature space, where samples from different classes are located near each other, leading to ambiguous decision boundaries and increased classification difficulty. A larger $R_{\mathrm{aug}}$ value corresponds to a higher

---
[1]URL: http://www.keel.es/

Table 2: Model performance on 44 imbalanced tabular datasets, shown as the average across datasets of 5-fold cross-validation results (**F1-score** on the left, **ROC-AUC** on the right). Corresponding standard deviations are reported in Appendix A.4. The best results are highlighted in **bold**. We compare plain MLP without graph structure, $GCN_{kNN}$ that leverages a fixed $k$-nearest neighbor graph, and various GSL models that dynamically learn the graph structure from data.

| | F1-score (↑) | | | | | | ROC-AUC (↑) | | | | | |
|---|---|---|---|---|---|---|---|---|---|---|---|---|
| Data sets | MLP | $GCN_{kNN}$ | SUBLIME | HES | IDGL | GILA | MLP | $GCN_{kNN}$ | SUBLIME | HES | IDGL | GILA |
| Ecoli034vs5 | 0.505 | 0.728 | 0.911 | **0.921** | 0.871 | **0.921** | 0.939 | 0.857 | 0.926 | **0.988** | 0.925 | 0.969 |
| Yeast2vs4 | 0.747 | 0.762 | **0.825** | 0.723 | 0.671 | 0.817 | 0.937 | 0.908 | 0.957 | 0.922 | 0.916 | **0.978** |
| Ecoli067vs35 | 0.135 | 0.706 | 0.658 | 0.697 | 0.769 | **0.825** | 0.521 | 0.817 | 0.924 | 0.913 | 0.933 | **0.960** |
| Ecoli0234vs5 | 0.738 | 0.807 | **0.900** | 0.854 | 0.855 | 0.865 | 0.956 | 0.878 | 0.945 | **0.989** | 0.953 | 0.988 |
| Glass015vs2 | 0.179 | 0.080 | 0.430 | 0.290 | 0.080 | **0.825** | 0.569 | 0.700 | 0.845 | 0.673 | 0.652 | **0.920** |
| Yeast0359vs78 | 0.300 | 0.406 | 0.429 | 0.529 | 0.420 | **0.816** | 0.524 | 0.812 | 0.800 | 0.769 | 0.789 | **0.929** |
| Yeast0256vs3789 | 0.545 | 0.643 | 0.487 | 0.633 | 0.522 | **0.773** | 0.868 | 0.833 | 0.697 | 0.839 | 0.834 | **0.981** |
| Yeast02579vs368 | 0.818 | 0.827 | 0.724 | **0.841** | 0.775 | 0.802 | 0.944 | 0.940 | 0.929 | 0.942 | 0.937 | **0.968** |
| Ecoli046vs5 | 0.731 | 0.712 | 0.766 | **0.876** | 0.864 | 0.863 | 0.901 | 0.846 | 0.942 | **0.966** | 0.919 | 0.965 |
| Ecoli01vs235 | 0.702 | 0.707 | **0.847** | 0.793 | 0.766 | 0.828 | 0.932 | 0.858 | 0.962 | 0.934 | 0.930 | **0.978** |
| Ecoli0267vs35 | 0.230 | 0.656 | 0.663 | 0.811 | 0.742 | **0.860** | 0.560 | 0.824 | 0.897 | 0.887 | 0.909 | **0.959** |
| Glass04vs5 | 0.550 | 0.767 | 0.614 | 0.933 | 0.933 | **0.960** | 0.920 | 0.982 | 0.918 | 0.981 | **1.000** | **1.000** |
| Ecoli0346vs5 | 0.784 | 0.688 | **0.927** | 0.921 | 0.871 | 0.886 | 0.819 | 0.886 | **0.994** | 0.986 | 0.915 | 0.993 |
| Ecoli0347vs56 | 0.825 | 0.833 | 0.772 | 0.835 | **0.915** | 0.838 | 0.957 | 0.919 | 0.959 | 0.973 | 0.965 | **0.982** |
| Yeast05679vs4 | 0.517 | 0.605 | 0.467 | 0.558 | 0.334 | **0.887** | 0.851 | 0.843 | 0.819 | 0.871 | 0.855 | **0.964** |
| Vowel0 | 0.961 | 0.994 | 0.586 | 0.985 | **1.000** | 0.963 | **1.000** | **1.000** | 0.955 | **1.000** | **1.000** | 0.998 |
| Ecoli067vs5 | 0.636 | 0.651 | 0.581 | 0.793 | 0.819 | **0.887** | 0.878 | 0.841 | 0.915 | 0.966 | 0.927 | **0.975** |
| Glass016vs2 | 0.162 | 0.050 | 0.336 | 0.231 | 0.000 | **0.800** | 0.537 | 0.420 | 0.832 | 0.725 | 0.713 | **0.958** |
| Ecoli0147vs2356 | 0.766 | 0.739 | 0.543 | **0.815** | 0.726 | 0.787 | 0.935 | 0.891 | 0.893 | 0.932 | 0.939 | **0.945** |
| Leddigit02456789vs1 | 0.797 | 0.782 | 0.658 | 0.818 | 0.761 | **0.922** | 0.963 | 0.950 | 0.919 | 0.960 | 0.949 | **0.972** |
| Ecoli01vs5 | 0.756 | 0.783 | 0.695 | 0.876 | 0.871 | **0.899** | 0.953 | 0.865 | 0.917 | **0.997** | 0.958 | 0.960 |
| Glass06vs5 | 0.179 | 0.600 | 0.437 | 0.867 | 0.933 | **1.000** | 0.777 | 0.948 | 0.804 | 0.964 | **1.000** | **1.000** |
| Glass0146vs2 | 0.153 | 0.256 | 0.375 | 0.210 | 0.000 | **0.905** | 0.518 | 0.791 | 0.749 | 0.694 | 0.638 | **0.995** |
| Glass2 | 0.147 | 0.222 | **0.369** | 0.177 | 0.000 | 0.080 | 0.478 | 0.717 | **0.852** | 0.756 | 0.640 | 0.626 |
| Ecoli0147vs56 | 0.776 | 0.753 | 0.655 | 0.814 | **0.883** | 0.855 | 0.919 | 0.903 | 0.943 | **0.969** | 0.953 | 0.957 |
| Cleveland0vs4 | 0.180 | 0.062 | 0.477 | 0.533 | 0.830 | **0.851** | 0.759 | 0.468 | 0.829 | 0.947 | **0.977** | 0.973 |
| Ecoli0146vs5 | 0.590 | 0.757 | 0.751 | 0.855 | **0.893** | 0.871 | 0.938 | 0.860 | 0.948 | **0.981** | 0.956 | 0.952 |
| Shuttlec0vsc4 | 0.983 | 0.992 | **1.000** | 0.996 | 0.996 | 0.996 | 0.992 | 0.998 | **1.000** | 0.992 | 0.995 | 0.997 |
| Yeast1vs7 | 0.237 | 0.294 | **0.365** | 0.216 | 0.230 | 0.202 | 0.712 | 0.749 | **0.807** | 0.659 | 0.739 | 0.704 |
| Glass4 | 0.080 | **0.831** | 0.468 | 0.638 | 0.767 | 0.743 | 0.646 | **0.989** | 0.907 | 0.914 | 0.940 | 0.931 |
| Ecoli4 | 0.651 | **0.893** | 0.840 | 0.886 | 0.861 | 0.883 | 0.991 | 0.975 | **0.995** | 0.956 | 0.981 | 0.978 |
| Page_blocks13vs4 | 0.557 | 0.869 | 0.858 | 0.848 | **0.933** | 0.904 | 0.818 | 0.991 | **0.995** | 0.993 | 0.982 | 0.977 |
| Abalone918 | **0.499** | 0.356 | 0.218 | 0.471 | 0.354 | 0.362 | **0.927** | 0.849 | 0.747 | 0.841 | 0.848 | 0.819 |
| Glass016vs5 | 0.206 | 0.467 | 0.465 | 0.733 | **0.960** | **0.960** | 0.854 | 0.786 | 0.915 | 0.940 | **1.000** | **1.000** |
| Shuttlec2vsc4 | 0.533 | 0.400 | **1.000** | **1.000** | 0.867 | 0.867 | **1.000** | **1.000** | **1.000** | **1.000** | **1.000** | 0.996 |
| Yeast1458vs7 | 0.078 | 0.038 | 0.178 | 0.076 | 0.000 | **0.799** | 0.283 | 0.555 | 0.595 | 0.567 | 0.593 | **0.900** |
| Glass5 | 0.090 | 0.439 | 0.344 | 0.733 | **0.960** | **0.960** | 0.839 | 0.910 | 0.881 | 0.883 | 0.998 | **1.000** |
| Yeast2vs8 | 0.601 | **0.690** | 0.623 | 0.646 | 0.668 | 0.668 | 0.722 | **0.859** | 0.825 | 0.747 | 0.802 | 0.786 |
| Yeast4 | 0.070 | **0.423** | 0.361 | 0.413 | 0.144 | 0.244 | 0.798 | 0.893 | **0.902** | 0.882 | 0.873 | 0.852 |
| Yeast1289vs7 | 0.051 | 0.079 | 0.194 | 0.055 | 0.000 | **0.669** | 0.471 | 0.449 | 0.773 | 0.704 | 0.646 | **0.922** |
| Yeast5 | 0.558 | 0.652 | 0.491 | **0.694** | 0.609 | 0.583 | **0.988** | 0.984 | 0.960 | 0.985 | 0.976 | 0.961 |
| Ecoli0137vs26 | 0.667 | 0.667 | 0.799 | 0.733 | 0.813 | **0.833** | 0.932 | 0.978 | **0.998** | 0.916 | 0.895 | 0.886 |
| Yeast6 | 0.050 | 0.666 | 0.353 | 0.609 | 0.525 | **0.691** | 0.844 | **0.949** | 0.934 | 0.934 | 0.929 | 0.903 |
| Abalone19 | 0.020 | 0.000 | **0.033** | 0.009 | 0.000 | 0.000 | 0.670 | 0.409 | **0.742** | 0.536 | 0.644 | 0.550 |
| Mean | 0.462 | 0.576 | 0.579 | 0.658 | 0.632 | **0.772** | 0.803 | 0.838 | 0.887 | 0.886 | 0.885 | **0.932** |
| Count | 1 | 4 | 9 | 6 | 7 | **21** | 4 | 5 | 10 | 8 | 6 | **21** |

degree of class overlap; its formal definition is provided in Appendix A.1. These datasets not only exhibit severe class imbalance but also present a high degree of class overlap, making them particularly challenging for classification tasks.

**Baseline** To evaluate the performance of GILA on classification tasks involving small-scale tabular imbalanced data, we compare it with two classical baselines and three GSL models. As baselines, we employ a Multi-Layer Perceptron (MLP) that does not utilize any structural information, and a GCN trained on a simple $k$-NN graph. The $k$-NN graph is constructed based on the similarity of node features, without any additional graph structure refinement. For GSL models, we utilize three representative methods—SUBLIME Liu et al. (2022), HES Wu et al. (2023), and IDGL Chen et al. (2020)—which have demonstrated strong performance in the GSL benchmark Li et al. (2023). These models are designed for learning instance-level graphs, where the graph is constructed among data instances.

**Experimental Setup** We conduct 5-fold cross-validation and represent the average of the results from the five trials. We use StratifiedKFold from scikit-learn to maintain the imbalance ratio in each fold. One fold out of the five folds is assigned for testing, and the rest are designated for training. To address concerns about potential bias in specific folds, we extract 25% of the entire training set at random for validation instead of using a single fold. Similarly, the stratify option is applied to ensure that the imbalance ratio in the validation set is maintained. Detailed implementation settings are provided in Appendix A.2. The search spaces of hyperparameters for all models are listed in Appendix A.3, and the final selected values are included in the supplementary material.

**Evaluation Metric** To validate the performance of the proposed method, we employ F1-score and ROC-AUC as evaluation metrics. F1-score is one of the widely used evaluation metrics for imbalanced data. Recall and Precision are in a trade-off relationship, and F1-score is their harmonic mean. While ROC-AUC is not sensitive to the distribution of two classes, the F1-score is sensitive to the accuracy of the minority class.

### 4.2 Performance Comparison with GSL Methods

We evaluate the classification performance on 44 small-scale and imbalanced tabular datasets by comparing the F1-score and ROC-AUC of our approach with those of existing GSL methods, as shown in Table 2. Our method achieves remarkable performance improvements, ranking first in F1-score on 21 out of 44 datasets and in ROC-AUC on 21 out of 44 datasets. Compared to HES, which is the best-performing GSL baseline method, GILA improves the mean F1-score across 44 datasets by 17.29% and the mean ROC-AUC by 5.22%, demonstrating significant enhancement in minority class classification. Graph-based baselines show improved performance over MLP, offering benefits by capturing instance-level dependencies. The superior performance of GSL models over $GCN_{kNN}$ further indicates that learning task-specific graph structures contributes to classification performance. Building on this advantage, GILA further outperforms all other methods by additionally alleviating class imbalance through the introduction of helper nodes.

### 4.3 Investigating the Effect of Class Overlap on the Performance of GSL Models

Class overlap—where samples share similar features with those from other class, leading to ambiguous decision boundaries—makes classification substantially more challenging. By analyzing the correlation between class overlap and classification performance, we observe that GILA is more robust to class overlap compared to other GSL methods. We attribute this robustness to the helper nodes in GILA, which are trained to promote class separation among connected samples—even when the features of the two classes are highly similar. Figure 3 illustrates the correlation between the degree of class overlap in each dataset and the F1-scores of the GSL models. Each point in the figure corresponds to a different dataset, with its class overlap ratio on the x-axis and the model's F1-score on the y-axis. While $GCN_{kNN}$, SUBLIME, HES, and IDGL exhibit strong negative correlations between F1-scores and class overlap (with coefficients of -0.76, -0.75, -0.73, and -0.65, respectively), GILA shows a noticeably weaker correlation of -0.40. This suggests that GILA's performance is less affected by class overlap.

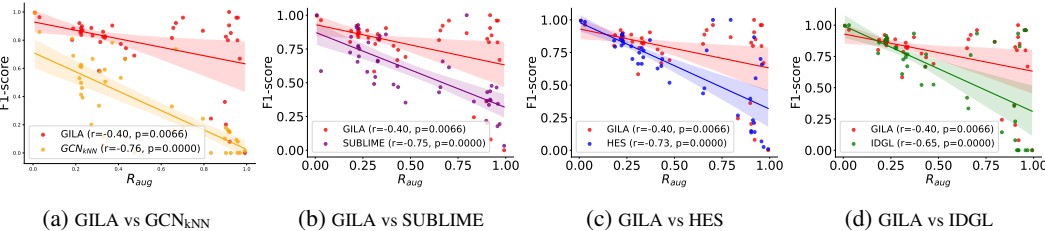

| (a) GILA vs $GCN_{kNN}$ | (b) GILA vs SUBLIME | (c) GILA vs HES | (d) GILA vs IDGL |

Figure 3: Correlation analysis between class overlap and GSL model performance. A higher $R_{aug}$ indicates greater class overlap. The observed weak correlation suggests robustness to overlapping classes.

## 4.4 COMPARISON GILA AGAINST GSL ENHANCED BY IMBALANCE TECHNIQUES: CLASS WEIGHTING AND OVERSAMPLING

Figure 4 shows the performance of GSL models when combined with commonly used class imbalance mitigation techniques, namely class weighting and oversampling (SMOTE Chawla et al. (2002)). The class weighting approach, which assigns higher penalties to misclassified instances of minority classes, proves relatively effective for MLP models. However, it results in a performance drop when applied to graph-based methods. In the oversampling approach, which balances class distribution by synthesizing new minority samples via interpolation, slight performance improvements were observed in GCN$_{kNN}$ and IDGL; however, the gains were marginal, and the other models performed worse than their vanilla counterparts. These results show that simply applying

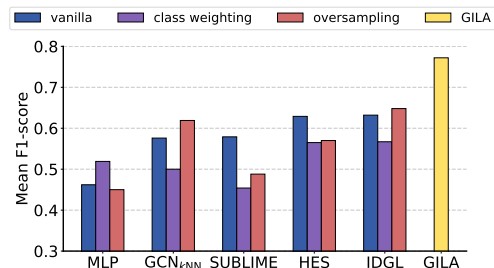

Figure 4: Comparison of GILA with class imbalance handling methods (Class Weighting and Oversampling). Results are mean F1-scores over 44 imbalanced tabular datasets.

imbalance-handling techniques to GSL models is inadequate, serving only as an ad-hoc remedy. In contrast, GILA consistently outperforms all other approaches by a substantial margin, jointly optimizing graph structures and enhancing node representation separability through helper nodes.

## 4.5 VISUALIZING THE IMPACT OF HELPER NODES ON NODE REPRESENTATIONS

We visually demonstrate how helper nodes affect the representation learning of their neighbors through message propagation. Figure 5 shows the evolution of node representations throughout the training process. We use t-distributed Stochastic Neighbor Embedding (t-SNE) Van der Maaten & Hinton (2008) for visualization. In (a), which depicts the initial state before training, samples from the minority and majority classes are heavily interwoven, and the helper nodes—still untrained—are positioned far from the existing samples. In (b), which represents an intermediate state of training, the helper nodes start to move closer to the minority class samples, contributing to their clustering. In (c), the helper nodes' representations are positioned between the minority and majority classes, playing a role in promoting class separation. By the end of training, as shown in (d), the representations of the minority and majority classes become completely distinct, demonstrating effective class separation.

Figure 6 presents graph visualizations after graph structure learning, comparing results without (a) and with (b) helper nodes. In the absence of helper nodes, applying GSL alone results in minority class samples being interspersed among majority class samples, resulting in overlapping representations across the two classes. In contrast, when helper nodes are introduced, minority class samples

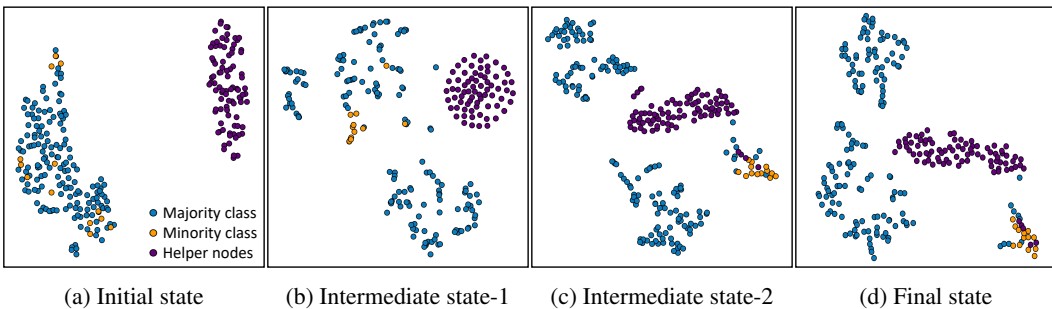

|(a) Initial state|(b) Intermediate state-1|(c) Intermediate state-2|(d) Final state|

Figure 5: Evolution of node representations during training on the Cleveland0vs4 dataset. To balance the class distribution (1:1 ratio), helper nodes are added to match the number of minority class samples to the majority class. Each subplot shows the distribution of node embeddings at different training states: (a) initial state before training, (b) and (c) intermediate states during training, and (d) final state after training.

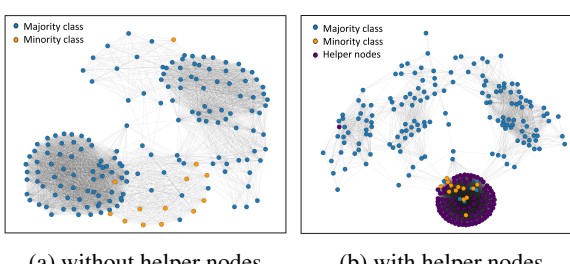 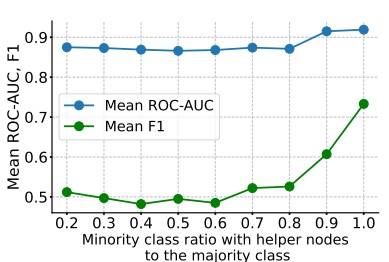

| (a) without helper nodes | (b) with helper nodes |

Figure 6: Graph visualization after graph structure learning without (a) and with (b) helper nodes on the Cleveland0vs4 dataset.

Figure 7: Effect of increasing helper nodes for the minority class on model performance.

### 4.6 QUANTIFYING THE EFFECT OF HELPER NODES ON CLASS SEPARATION

To quantitatively assess how helper nodes improve class separability, we measure the margin between the minority and majority classes in the embedding space. Specifically, we classify the final node embeddings using a linear SVM and measure the resulting decision margins. The evaluation is performed across all 44 tabular datasets. As shown in Table 3, incorporating helper nodes consistently improves both accuracy and the inter-class decision margin. These results further support our claim that helper nodes help produce more separable class representations in the embedding space.

Table 3: Effect of helper nodes on SVM-based inter-class margins. Results are averaged over all 44 datasets.

| Setting | Mean Accuracy | Mean Margin |
|---|---|---|
| w/o helper node | 0.858 | 1.995 |
| w helper node | **0.902** | **2.604** |

### 4.7 EFFECT OF INCREASING HELPER NODES FOR MINORITY CLASS

Figure 7 illustrates the impact of increasing the number of helper nodes for the minority class. A ratio of 1 indicates that enough helper nodes are added to make the number of minority nodes equal to that of the majority class (only in the training set). The F1-score increases sharply when the ratio exceeds 0.8, with the best performance observed at full balance. While the ROC-AUC remains relatively stable, a slight improvement is seen from a ratio of 0.8. These results suggest that balancing the training graph by adding helper nodes is an effective strategy.

## 5 CONCLUSION

We propose a simple yet powerful graph structure learning approach that is effective for small-scale, class-imbalanced tabular datasets. We model the latent relationships between instances using GSL and introduce helper nodes for the minority class, which are optimized in the feature space by the updater. These helper nodes promote the separability of the minority class from the majority class through message propagation. As a result, the proposed method outperforms the baseline across 44 datasets. Even under class overlap—where distinguishing between classes is inherently difficult—GILA demonstrates remarkable robustness, primarily due to the helper nodes that promote representation-level separation. Representation visualizations illustrate how helper nodes progressively drive the separation of class representations during training, while graph visualizations confirm that they induce clearly separated subgraphs for minority and majority classes. Nonetheless, this study has a limitation. As the proposed method operates in a transductive setting, it cannot directly accommodate newly added data. Future work will focus on extending the method to inductive scenarios and exploring other challenging settings.

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

# A    APPENDIX

## A.1    ANALYSIS OF CLASS OVERLAP IN THE DATASET

The degree of class overlap in imbalanced data was measured using the augmented R-value ($R_{\text{aug}}$) Fu et al. (2020), where a higher $R_{\text{aug}}$ indicates a greater proportion of class overlap. Given a binary classification dataset $D$ with two classes, $R_{\text{aug}}$ is computed as follows:

- Let $C_i$ be the set of instances belonging to class $i$, and $P_m^i$ denote the $m$-th instance in $C_i$.
- For each instance $P_m^i$, identify its $k$-nearest neighbors from the entire dataset $U$, denoted as $\text{kNN}(P_m^i, U)$.
- Count the number of neighbors that belong to the complement of $C_i$, i.e., $\complement(C_i)$.
- Define a threshold $\theta$. If the number of neighbors from $\complement(C_i)$ exceeds $\tau$, consider $P_m^i$ to lie in an overlapping region.
- The R-value of class $C_i$ is defined as:

$$R(C_i) = \frac{1}{|C_i|} \sum_{m=1}^{|C_i|} \text{sign}\left(|\text{kNN}(P_m^i, \complement(C_i))| - \tau\right), \tag{9}$$

   where

$$\text{sign}(x) = \begin{cases} 1 & \text{if } x > 0, \\ 0 & \text{otherwise.} \end{cases}$$

- Let $C_0$ and $C_1$ be the sets of instances belonging to the majority and minority classes, respectively.
- The class imbalance ratio $r$ is defined as:

$$r = \frac{|C_0|}{|C_1|}. \tag{10}$$

- The Augmented R-value is then computed by:

$$R_{\text{aug}}(D) = \frac{R(C_0) + r \times R(C_1)}{r + 1}. \tag{11}$$

Table 4: Hyperparameter search space for all models

| Model | Hyperparameter | Search Space |
|---|---|---|
| MLP | Learning rate
Hidden dim | [0.001]
[32, 64, 128, 256] |
| $GCN_{kNN}$ | Learning rate
Hidden dim
$k$-NN size | [0.01]
[32, 64, 128, 256]
[5, 10, 20, 30, 40,
50, 60, 70, 80, 90, 100] |
| SUBLIME | Type learner
$k$-NN size
Hidden dim | MLP
[5, 10, 20, 30, 40,
50, 60, 70, 80, 90, 100]
[32, 64, 128, 256] |
| HES | Learning rate
$k$-NN size
Cls Hidden dim
DAE Hidden dim | [0.001]
[5, 10, 20, 30, 40,
50, 60, 70, 80, 90, 100]
[32, 64, 128, 256]
[32, 64, 128, 256] |
| IDGL | Learning rate
Init. adj. ratio ($\lambda$)
Update adj. ratio ($\eta$)
Hidden dim
$k$-NN size | [0.01]
[0.7]
[0, 0.5, 0.6, 0.7, 0.8, 0.9]
[16, 32, 64]
[5, 10, 20, 30, 40,
50, 60, 70, 80, 90, 100] |
| **GILA** | Learning rate
Init. adj. ratio ($\lambda$)
Update adj. ratio ($\eta$)
Hidden dim
$k$-NN size | [0.01]
[0.7]
[0.5, 0.6, 0.7, 0.8, 0.9]
[16, 32, 64]
[5, 10, 20, 30, 40,
50, 60, 70, 80, 90, 100] |

## A.2 IMPLEMENTATION DETAILS

All experiments involving GILA were conducted using PyTorch 1.13.1 with CUDA 11.6 and cuDNN 8. The hardware setup consisted of a single NVIDIA RTX 3090 GPU, an Intel(R) Core(TM) i9-10900X CPU @ 3.70GHz, and 128GB of RAM. The operating system used was Ubuntu 18.04.6 LTS. For baseline models, we used their official implementations and default environments whenever possible, as provided by the original authors or repositories.

## A.3 HYPERPARAMETER SEARCH SPACE

To ensure a fair and optimal comparison, we performed hyperparameter tuning separately on each of the 44 datasets. For every dataset, we selected the best-performing configuration on the validation set. Although only key hyperparameters were tuned, all remaining parameters were kept at default values based on official implementations. In GILA, the structure learner is implemented using IDGL, as described in the main text. The architecture is exactly the same as the IDGL model used as a baseline. The hyperparameter search spaces used for each model are summarized in Table 4.

## A.4 STANDARD DEVIATIONS OF MAIN RESULTS

Table 5 reports the average standard deviations of the F1-score and ROC-AUC computed from 5-fold cross-validation within each dataset, across the 44 imbalanced tabular datasets for each model. These values correspond to the results shown in Table 2 and offer insights into the stability and consistency of model performance across folds.

## A.5 COMPARISON WITH EXISTING IMBALANCED GNN METHODS

Table 5: **Standard deviations (Std.)** of F1-score (left) and ROC-AUC (right) across 44 imbalanced tabular datasets. Each value reflects the standard deviation across datasets of mean performance scores obtained via 5-fold cross-validation, and corresponds to the average results reported in Table 2.

| Data sets | Std. of F1-score | | | | | | Std. of ROC-AUC | | | | | |
|---|---|---|---|---|---|---|---|---|---|---|---|---|
| | MLP | GCN$_{kNN}$ | SUBLIME | HES | IDGL | GILA | MLP | GCN$_{kNN}$ | SUBLIME | HES | IDGL | GILA |
| Ecoli034vs5 | 0.080 | 0.180 | 0.028 | 0.102 | 0.080 | 0.087 | 0.064 | 0.092 | 0.023 | 0.022 | 0.077 | 0.054 |
| Yeast2vs4 | 0.102 | 0.125 | 0.022 | 0.113 | 0.104 | 0.091 | 0.028 | 0.040 | 0.012 | 0.055 | 0.041 | 0.019 |
| Ecoli067vs35 | 0.124 | 0.175 | 0.125 | 0.219 | 0.149 | 0.176 | 0.083 | 0.177 | 0.041 | 0.128 | 0.077 | 0.057 |
| Ecoli0234vs5 | 0.173 | 0.142 | 0.035 | 0.107 | 0.133 | 0.170 | 0.047 | 0.092 | 0.026 | 0.010 | 0.048 | 0.025 |
| Glass015vs2 | 0.021 | 0.160 | 0.032 | 0.289 | 0.160 | 0.108 | 0.119 | 0.093 | 0.008 | 0.171 | 0.118 | 0.079 |
| Yeast0359vs78 | 0.181 | 0.082 | 0.013 | 0.156 | 0.148 | 0.054 | 0.082 | 0.030 | 0.013 | 0.082 | 0.061 | 0.041 |
| Yeast0256vs3789 | 0.083 | 0.080 | 0.017 | 0.075 | 0.037 | 0.088 | 0.041 | 0.077 | 0.017 | 0.043 | 0.034 | 0.010 |
| Yeast02579vs368 | 0.046 | 0.066 | 0.017 | 0.065 | 0.082 | 0.072 | 0.040 | 0.043 | 0.013 | 0.035 | 0.032 | 0.019 |
| Ecoli046vs5 | 0.146 | 0.159 | 0.038 | 0.123 | 0.080 | 0.157 | 0.082 | 0.121 | 0.023 | 0.037 | 0.082 | 0.057 |
| Ecoli01vs235 | 0.120 | 0.200 | 0.065 | 0.112 | 0.241 | 0.119 | 0.079 | 0.124 | 0.022 | 0.059 | 0.037 | 0.013 |
| Ecoli0267vs35 | 0.303 | 0.205 | 0.158 | 0.198 | 0.118 | 0.215 | 0.256 | 0.148 | 0.066 | 0.141 | 0.090 | 0.068 |
| Glass04vs5 | 0.174 | 0.200 | 0.030 | 0.133 | 0.133 | 0.080 | 0.095 | 0.015 | 0.029 | 0.038 | 0.000 | 0.000 |
| Ecoli0346vs5 | 0.425 | 0.117 | 0.051 | 0.102 | 0.112 | 0.102 | 0.214 | 0.074 | 0.008 | 0.017 | 0.087 | 0.010 |
| Ecoli0347vs56 | 0.178 | 0.078 | 0.106 | 0.113 | 0.043 | 0.069 | 0.051 | 0.089 | 0.015 | 0.023 | 0.046 | 0.021 |
| Yeast05679vs4 | 0.067 | 0.157 | 0.044 | 0.108 | 0.112 | 0.071 | 0.092 | 0.089 | 0.013 | 0.062 | 0.055 | 0.030 |
| Vowel0 | 0.023 | 0.011 | 0.016 | 0.031 | 0.000 | 0.038 | 0.001 | 0.000 | 0.010 | 0.001 | 0.000 | 0.003 |
| Ecoli067vs5 | 0.205 | 0.242 | 0.100 | 0.052 | 0.076 | 0.065 | 0.095 | 0.128 | 0.033 | 0.024 | 0.085 | 0.047 |
| Glass016vs2 | 0.019 | 0.100 | 0.055 | 0.159 | 0.000 | 0.245 | 0.143 | 0.148 | 0.034 | 0.158 | 0.070 | 0.057 |
| Ecoli0147vs2356 | 0.083 | 0.089 | 0.012 | 0.097 | 0.086 | 0.067 | 0.057 | 0.091 | 0.007 | 0.077 | 0.086 | 0.080 |
| Leddigit02456789vs1 | 0.094 | 0.094 | 0.110 | 0.088 | 0.057 | 0.077 | 0.022 | 0.038 | 0.032 | 0.023 | 0.026 | 0.037 |
| Ecoli01vs5 | 0.207 | 0.081 | 0.189 | 0.123 | 0.080 | 0.094 | 0.054 | 0.111 | 0.044 | 0.003 | 0.060 | 0.074 |
| Glass06vs5 | 0.097 | 0.327 | 0.013 | 0.163 | 0.133 | 0.000 | 0.055 | 0.034 | 0.024 | 0.044 | 0.000 | 0.000 |
| Glass0146vs2 | 0.020 | 0.135 | 0.114 | 0.310 | 0.000 | 0.131 | 0.118 | 0.109 | 0.096 | 0.182 | 0.130 | 0.007 |
| Glass2 | 0.019 | 0.200 | 0.049 | 0.160 | 0.000 | 0.160 | 0.179 | 0.138 | 0.061 | 0.175 | 0.161 | 0.198 |
| Ecoli0147vs56 | 0.108 | 0.130 | 0.079 | 0.116 | 0.088 | 0.097 | 0.071 | 0.080 | 0.038 | 0.043 | 0.057 | 0.056 |
| Cleveland0vs4 | 0.223 | 0.123 | 0.122 | 0.267 | 0.087 | 0.078 | 0.222 | 0.144 | 0.057 | 0.062 | 0.012 | 0.016 |
| Ecoli0146vs5 | 0.315 | 0.104 | 0.072 | 0.133 | 0.096 | 0.112 | 0.065 | 0.111 | 0.014 | 0.024 | 0.062 | 0.073 |
| Shuttlec0vsc4 | 0.016 | 0.010 | 0.000 | 0.009 | 0.008 | 0.008 | 0.016 | 0.004 | 0.000 | 0.017 | 0.010 | 0.007 |
| Yeast1vs7 | 0.059 | 0.139 | 0.071 | 0.166 | 0.200 | 0.189 | 0.088 | 0.121 | 0.046 | 0.117 | 0.105 | 0.125 |
| Glass4 | 0.072 | 0.183 | 0.103 | 0.132 | 0.200 | 0.164 | 0.206 | 0.010 | 0.065 | 0.151 | 0.090 | 0.091 |
| Ecoli4 | 0.146 | 0.096 | 0.151 | 0.057 | 0.134 | 0.122 | 0.010 | 0.036 | 0.005 | 0.072 | 0.025 | 0.026 |
| Page_blocks13vs4 | 0.197 | 0.055 | 0.060 | 0.128 | 0.063 | 0.083 | 0.115 | 0.006 | 0.002 | 0.006 | 0.032 | 0.043 |
| Abalone918 | 0.214 | 0.129 | 0.028 | 0.107 | 0.093 | 0.175 | 0.058 | 0.080 | 0.039 | 0.086 | 0.085 | 0.113 |
| Glass016vs5 | 0.123 | 0.400 | 0.174 | 0.389 | 0.080 | 0.080 | 0.063 | 0.340 | 0.055 | 0.106 | 0.000 | 0.000 |
| Shuttlec2vsc4 | 0.452 | 0.490 | 0.000 | 0.000 | 0.163 | 0.163 | 0.000 | 0.000 | 0.000 | 0.000 | 0.000 | 0.008 |
| Yeast1458vs7 | 0.007 | 0.076 | 0.048 | 0.065 | 0.000 | 0.112 | 0.035 | 0.130 | 0.084 | 0.185 | 0.101 | 0.065 |
| Glass5 | 0.111 | 0.247 | 0.044 | 0.389 | 0.080 | 0.080 | 0.050 | 0.056 | 0.034 | 0.234 | 0.005 | 0.000 |
| Yeast2vs8 | 0.208 | 0.168 | 0.037 | 0.244 | 0.151 | 0.151 | 0.185 | 0.046 | 0.017 | 0.245 | 0.088 | 0.089 |
| Yeast4 | 0.085 | 0.145 | 0.020 | 0.104 | 0.147 | 0.130 | 0.133 | 0.027 | 0.013 | 0.042 | 0.021 | 0.016 |
| Yeast1289vs7 | 0.026 | 0.112 | 0.060 | 0.068 | 0.000 | 0.088 | 0.129 | 0.227 | 0.027 | 0.103 | 0.041 | 0.053 |
| Yeast5 | 0.206 | 0.129 | 0.021 | 0.143 | 0.089 | 0.079 | 0.005 | 0.009 | 0.010 | 0.003 | 0.013 | 0.014 |
| Ecoli0137vs26 | 0.365 | 0.365 | 0.112 | 0.389 | 0.244 | 0.211 | 0.088 | 0.028 | 0.001 | 0.167 | 0.161 | 0.179 |
| Yeast6 | 0.100 | 0.116 | 0.033 | 0.156 | 0.164 | 0.097 | 0.050 | 0.026 | 0.021 | 0.044 | 0.042 | 0.046 |
| Abalone19 | 0.004 | 0.000 | 0.006 | 0.011 | 0.000 | 0.000 | 0.104 | 0.071 | 0.037 | 0.249 | 0.125 | 0.089 |
| Mean | 0.137 | 0.150 | 0.061 | 0.142 | 0.097 | 0.108 | 0.086 | 0.083 | 0.028 | 0.081 | 0.059 | 0.048 |

We evaluate existing imbalanced graph learning methods on tabular data by constructing a kNN graph—a common and straightforward approach for adapting non-graph data to graph models. The results across the 44 datasets are reported in Table 6. Our method achieves substantially higher performance in this setting, indicating that a kNN graph is not an optimal structure for tabular data. By jointly learning the task-specific graph structure and the helper nodes, GILA delivers significantly better performance.

Table 6: Comparison with existing imbalanced GNN methods on kNN-constructed graphs (Mean F1 across 44 datasets).

| Method | Mean F1 |
|---|---|
| GraphSMOTE (kNN) | 0.681 |
| GraphENS (kNN) | 0.655 |
| **GILA (ours)** | **0.772** |

## A.6 LLM USAGE

We used a large language model (ChatGPT) only for minor grammar correction and polishing of the writing. The LLM was not involved in research ideation, experimental design, analysis, or interpretation of results.

