# OpenReview forum: "GILA: Graph Structure Learning for Class-Imbalanced Tabular Data"
_ICLR.cc/2026/Conference — Submitted to ICLR 2026_

### Official Review · Reviewer_e9XP · 2025-10-16

**Soundness:** 2
**Presentation:** 2
**Contribution:** 1
**Rating:** 2
**Confidence:** 3

**Summary:**

This paper proposes GILA, a graph structure learning approach for class-imbalanced tabular data that introduces helper nodes to assist minority class classification. Extensive experimental evaluation is conducted across 44 imbalanced datasets.

**Strengths:**

1. Extensive experimental evaluation across 44 imbalanced datasets

**Weaknesses:**

1. Limited Technical Novelty: The core contribution—helper nodes with learnable parameters—is essentially a form of synthetic minority augmentation in the imbalanced classification domain. While the updater mechanism (Eq. 5-6) adds learnable parameters p_i to zero-initialized helper nodes, this is conceptually similar to existing synthetic oversampling techniques (SMOTE). The paper doesn't sufficiently justify why this graph-based approach is superior to simpler alternatives.


2. Incomplete Baseline Comparison: Given that the evaluation focuses on tabular data, the absence of standard tabular ML baselines is an omission. The paper should include the following baselines: Gradient boosting methods (XGBoost, LightGBM); Ensemble methods (Random Forest, AdaBoost). These methods need to be equipped with SMOTE/class weighting for fair comparison. Without these baselines, it's unclear whether the graph structure learning paradigm actually provides benefits over established tabular methods.

3. Insufficient Technical Details: The graph regularization loss L_G (mentioned in Section 3.3) is referenced but not formally defined. Also, the complexity analysis is missing—both time and space complexity should be analyzed, especially given the addition of helper nodes. How does computational complexity scale with the number of helper nodes? Moreover, algorithm or pseudocode for the complete training procedure would help improve clarity.

4. Performance Degradation in Extreme Imbalance: The model completely fails on Abalone19 (IR=129.44), achieving F1=0.000 and ROC-AUC=0.5, which is equivalent to random guessing. This failure needs explanation or discussion.

**Questions:**

See Weakness.

---

> ### Author Response · Authors · 2025-11-20
>
> We appreciate the reviewer’s comments.
>
> We would like to clarify a fundamental misunderstanding regarding the problem setting of our work.
>
> **(1) Difference in problem formulation.**
>
> There is a fundamental difference in the problem formulation.
> Existing imbalanced graph node classification assumes that a **graph is already provided**, and minority nodes are supplemented through resampling or synthetic generation, after which the generated nodes are connected to the existing graph structure.
> In contrast, our method handles tabular data with **no given graph**, and the model must construct the entire graph structure from scratch , reflecting the underlying relationships among samples.
> To avoid any possible confusion, we will add a brief clarification in the problem definition part.
>
> **(2) Difference in the role of additional nodes.**
>
> In prior work, synthetic nodes are treated as data samples—typically generated by interpolating existing minority features or modeling their distribution—and then inserted into the graph as new minority instances.
> However, our helper nodes are **not data points**. They do not learn or approximate the feature distribution of minority samples. Instead, they are connected to nodes from both classes and are trained to make their embeddings more separable in the embedding space. To support this claim, we provide additional experimental results.
>
> **(3) Supporting empirical evidence.**
>
> To further validate the effect of helper nodes, we provide an additional experiment where final node embeddings are classified using a linear SVM, and the SVM margin is measured.
> The following results show the average performance across all 44 datasets.
>
> | Setting             | Accuracy | Mean Margin |
> |---------------------|:----------:|:-------------:|
> | w/o helper node     | 0.858    | 1.995       |
> | w helper node    | 0.902    | 2.604       |
>
> As shown, adding helper nodes significantly increases the inter-class margin. This supports our hypothesis that helper nodes influence the message-passing aggregation process such that the embeddings of different classes become more separable.
>
> **(4) Comparison with Existing Imbalanced GNN Methods**
>
> Additionally, we tested whether existing imbalanced graph learning mehods can be applied to tabular data by constructing a kNN graph—a common and straightforward approach for adapting non-graph data to graph models. The results on the 44 datasets are as follows:
>
> | Method  | Mean F1 |
> |--------------------------|---------|
> | GraphSMOTE (kNN)         | 0.681   |
> | GraphENS (kNN)           | 0.655   |
> | **GILA (ours)**          | **0.772** |
>
> Our method achieves substantially higher performance in these experiments. This demonstrates that the kNN graph is not an optimal structure for tabular data. GILA learns both the task-specific graph structure and the helper nodes, resulting in significantly better performance.
>
>
> **Continued in the next comment.**

---

> ### Author Response · Authors · 2025-11-20
>
> **(5)  Comparison with ML baseline**
>
> We have already evaluated standard ML models, and we present the results here. The table below reports the performance of ML baselines as well as their class-weighted (cw) and resampling variants (mean F1-score over the 44 datasets). Here, RF denotes Random Forest, and BSMOTE refers to Borderline-SMOTE.
> CDBH and SSHR are the models provided in the dataset sources we follow.
> Our method significantly outperforms all ML baselines and their resampling/class-weighted variants.
>
>
> | Data sets | RF | GBM | XGB | RF (cw) | GBM (cw) | XGB (cw) | SMOTE | BSMOTE | CDBH | SSHR | GILA |
> |-----------|:------:|:------:|:------:|:------:|:------:|:------:|:------:|:------:|:------:|:------:|:------:|
> | **Mean** | 0.570 | 0.598 | 0.594 | 0.556 | 0.598 | 0.594 | 0.641 | 0.619 | 0.598 | 0.644 | **0.772** |
> | **Count** | 1 | 3 | 0 | 1 | 0 | 0 | 3 | 4 | 2 | 3 | 27 |
>
>
> **(6) Clarifications on Graph Regularization Loss**
>
> We would like to clarify that the graph regularization loss L_g is not proposed by our method, but is inherited directly from IDGL. As described in Section 2.3 (“IDGL”) of the paper, we briefly introduced this loss only to provide the necessary background, since it is part of the original IDGL framework that we build upon. For clarity, we will explicitly state in the revised version that L_g is entirely adopted from IDGL without modification, and that its role and formulation were already established in the original work.
>
> **(7) Complexity Analysis with Helper Nodes**
>
> To address the reviewer’s concern regarding the computational and memory cost introduced by helper nodes, we provide a detailed complexity analysis below.
>
> Let **N** denote the number of original nodes, **M** the number of helper nodes, and **d** the feature dimension (shared by both original and helper nodes).
>
> After adding **M** helper nodes, the total number of nodes becomes **N + M**.
> The complexity of learning the adjacency matrix changes from **O(N² d)** to **O((N + M)² d)**, which expands to:
>
> (N + M)² d = (N² + 2NM + M²) d.
>
> In our setting, helper nodes are introduced only for the minority class to roughly balance the majority and minority classes.
> Since the minority class is never empty, the number of helper nodes **M** is strictly smaller than the original number of nodes, i.e., **M < N**.
> This implies:
>
> N < N + M < 2N,
>
> and thus the adjacency-learning complexity is upper bounded by:
>
> O((N + M)² d) < O((2N)² d) = O(4 N² d).
>
> That is, our method increases the cost by at most a constant factor (strictly less than 4× in this setting), while preserving the same asymptotic order **O(N² d)** as the original IDGL.
>
> The helper nodes are learnable, but updating their embeddings only costs **O(M d)**, which is insignificant relative to the adjacency-learning cost **O((N + M)² d)**.
>
> Therefore, the overall computational complexity remains of the same order as the original IDGL, with a moderate constant-factor overhead, while providing improved performance on imbalanced classes.
>
> (8) **Discussion on Specific Datasets**
>
> To maintain **fairness and transparency**, we report all results—including challenging datasets such as Abalone19—rather than selectively excluding difficult cases.
>
> Abalone19 is extremely imbalanced (IR = 129.44), and we found that most of the comparison models fail to train effectively on this dataset. In such an extreme setting, traditional ML methods might be more appropriate in some cases. We acknowledge that this represents a limitation of our method.
>
> However, it is important to note that this dataset is a rare case. Across the vast majority of the 44 datasets, our method consistently outperforms deep-learning-based GSL approaches. Thus, while Abalone19 represents a situation where GILA is less effective, the overall results demonstrate that GILA achieves superior performance across a wide range of imbalanced tabular tasks.
>
> Thank you again for your valuable feedback.
> We hope our responses address your concerns, and we remain open to further clarification during the discussion phase.

---

### Official Review · Reviewer_Sjt8 · 2025-10-29

**Soundness:** 1
**Presentation:** 2
**Contribution:** 1
**Rating:** 2
**Confidence:** 4

**Summary:**

This paper proposes a Graph Structure Learning (GSL) framework named GILA, designed to address two challenging scenarios: small-scale datasets and class imbalance. The method learns latent inter-sample relationships and introduces auxiliary (helper) nodes to mitigate class imbalance, thereby improving classification performance on imbalanced tabular data. The authors validate their approach on 44 real-world imbalanced tabular datasets and provide visualizations to illustrate the role of auxiliary nodes.

**Strengths:**

- The paper addresses class imbalance in small-scale tabular data—a problem that has received limited attention in the graph structure learning literature—making it practically relevant.
- Explicitly modeling inter-sample relationships via graph structure learning to enhance few-shot learning capabilities is a reasonable and well-motivated approach.
- Extensive experiments on numerous real-world datasets are conducted, accompanied by visualizations that help explain the behavior of auxiliary nodes in the representation space.

**Weaknesses:**

1. The core innovation—introducing auxiliary nodes for minority classes and jointly training them to alleviate class imbalance—is highly similar to prior strategies such as GraphSMOTE, which rely on oversampling or virtual node generation. Although Section 4.4 includes comparative experiments, the paper lacks sufficient theoretical or mechanistic novelty to justify a significant technical advance. The current contribution appears intuitive rather than groundbreaking.

2. While the authors claim that GILA can capture underlying data distributions even with scarce samples, small-scale datasets inherently suffer from limited information and high noise sensitivity. Introducing auxiliary nodes may exacerbate noise propagation. The paper does not discuss the sensitivity of auxiliary node initialization or training dynamics, which could critically affect performance in minority-class learning.

3. The graph structure regularization loss \( L_g \) is not clearly defined or explained. Key hyperparameters (e.g., the threshold \( \epsilon \)) are inadequately discussed, making reproduction difficult.

4. The paper does not sufficiently address whether the construction and optimization of auxiliary nodes might introduce noise or lead to structural overfitting. No robustness analyses or ablation studies are provided to evaluate this risk.

5. On datasets such as Glass2 and Abalone19, GILA performs significantly worse than baseline methods. The authors should explain why performance varies across datasets with different scales and imbalance ratios. Moreover, the set of compared baselines is limited, especially lacking comparisons with recent imbalanced graph learning methods (e.g., ImGCL, GraphENS, G2GNN, ImbGNN). Additionally, results are reported only as averages without standard deviations, making it impossible to assess result stability.

6. In extremely imbalanced scenarios, GILA may require a large number of auxiliary nodes to balance class distributions, substantially increasing computational and memory overhead. The authors should provide a complexity analysis or propose a lightweight variant.

7.  The paper only employs GCN as the base classifier. It remains unclear whether GILA is compatible with other GNN architectures (e.g., GAT, GraphSAGE, GIN, or graph transformers) and how such choices affect performance.

8. Ambiguities in Visualization and Class Definition:
   - In Figure 4.5, during intermediate training stages (subfigures b–d), auxiliary nodes appear equidistant from both minority and majority classes, showing no clear separation. If auxiliary nodes are labeled as minority class, their final embeddings should align more closely with the minority distribution—this discrepancy requires clarification.
   - For multi-class settings, the paper does not specify how “majority” and “minority” classes are defined. Is the majority class the one with the most samples? Are minority classes defined as only the least frequent class, or as all non-majority classes?
     - If the latter, there could be many minority classes, raising questions about how many auxiliary nodes to introduce and how to train them.
     - If only the single rarest class receives auxiliary nodes, how are other low-frequency (but not the rarest) classes effectively learned?

9. The effectiveness of GILA is supported solely by empirical results, with no theoretical analysis (e.g., how auxiliary nodes influence graph smoothness, feature distribution, or decision boundaries). Given the notable performance fluctuations across datasets, the current evidence is insufficient to establish the method’s general validity or robustness in imbalanced learning scenarios.

**Questions:**

Please refer to the weakness.

---

> ### Author Response · Authors · 2025-11-20
>
> We appreciate the reviewer’s comments.
>
> We would like to clarify a fundamental misunderstanding regarding the problem setting of our work.
>
> **(1) Difference in problem formulation.**
>
> There is a fundamental difference in the problem formulation.
> Existing imbalanced graph node classification assumes that a **graph is already provided**, and minority nodes are supplemented through resampling or synthetic generation, after which the generated nodes are connected to the existing graph structure.
> In contrast, our method handles tabular data with **no given graph**, and the model must construct the entire graph structure from scratch , reflecting the underlying relationships among samples.
> To avoid any possible confusion, we will add a brief clarification in the problem definition part.
>
> **(2) Difference in the role of additional nodes.**
>
> In prior work, synthetic nodes are treated as data samples—typically generated by interpolating existing minority features or modeling their distribution—and then inserted into the graph as new minority instances.
> However, our helper nodes are **not data points**. They do not learn or approximate the feature distribution of minority samples. Instead, they are connected to nodes from both classes and are trained to make their embeddings more separable in the embedding space. To support this claim, we provide additional experimental results.
>
> **(3) On Distribution Estimation Concerns**
>
> As you correctly pointed out, estimating data distributions becomes extremely challenging when the dataset is small. However, as explained in Response 1, our method does **not attempt to estimate the data distribution.** This is explicitly stated in Lines 63–66 of the paper:
> “However, in many real-world tabular applications, data is often too scarce
> to support reliable distribution estimation. Instead of relying on distribution estimation, we leverage GSL to extract richer inter-sample information, enabling more effective learning from small-scale tabular data.”
> We introduce graph structure learning specifically to exploit inter-sample information, rather than to model the underlying data distribution. Furthermore, the helper nodes are not data samples. Instead, they influence the representation learning of neighboring nodes and help make the minority and majority class embeddings more separable during message passing.
>
> **(4) Clarifications on Graph Regularization Loss and Hyperparameters**
>
> We would like to clarify that the graph regularization loss L_g is not proposed by our method, but is inherited directly from IDGL. As described in Section 2.3 (“IDGL”) of the paper, we briefly introduced this loss only to provide the necessary background, since it is part of the original IDGL framework that we build upon. For clarity, we will explicitly state in the revised version that L_g is entirely adopted from IDGL without modification, and that its role and formulation were already established in the original work.
> As noted in Lines 330–331, the corresponding hyperparameter search space is provided in the Appendix to ensure reproducibility.
>
> **(5) On Noise and Structural Overfitting Concerns**
> As described in the Method section, GILA is trained through the co-training of IDGL and the helper nodes. In Section 4.2 and Table 2, we provide a direct comparison between IDGL alone and GILA (IDGL + helper nodes). The results show that GILA achieves substantial performance improvements over IDGL, which indicates that the helper nodes do not act as noise but instead make a meaningful and positive contribution to the learning process. To make this point clearer, we will add an explicit explanation in Section 4.2 in the revised version .
> Moreover, our method demonstrates strong performance on the majority of the 44 datasets, which we believe provides sufficient evidence that the approach does not suffer from structural overfitting. Validating the model across 44 tabular datasets with varying sizes and characteristics provides evidence that the method does not overfit to a specific dataset configuration. The consistent improvements observed across many datasets suggest that helper nodes do not introduce structural overfitting.
>
> **Continued in the next comment.**

---

> ### Author Response · Authors · 2025-11-20
>
> **(6) Performance Variance Across Datasets (Glass2 & Abalone19)**
>
> For small-scale imbalanced datasets, issues such as class overlap, noise, or limited sample diversity often coexist with class imbalance and significantly affect the results. Please refer to Table 2 in the paper (sorted by IR in ascending order). For example, Glass2 has an IR of 11.59, yet several datasets with much higher IRs are successfully learned by our model. This indicates that Glass2 is a more challenging dataset with difficulties beyond imbalance, such as severe class overlap.
> In the case of Abalone19, most comparison models also failed to learn effectively, which highlights a limitation of our approach as well. Importantly, we chose to report all results, including such difficult cases, to ensure fairness and transparency in evaluation. We believe that openly presenting both the strengths and limitations of our method is an essential part of a rigorous and honest scientific contribution.
>
> **(7) Comparison with Imbalanced GNN Methods**
>
> Regarding the request for additional baselines, as explained in Response (1), existing imbalanced graph learning methods are originally designed for graph-structured data. The most straightforward way to apply them to tabular data is to construct a kNN graph and use it as input. Accordingly, we conducted experiments with GraphSMOTE and GraphENS under this setting.
> Among the methods you suggested, **G2GNN** and **ImbGNN** are designed for **graph-level tasks**, and therefore are not directly comparable to our node-level tabular setting. Additionally, **ImGCL** does not provide publicly available code, which makes reliable reproduction infeasible.
> The results on the 44 datasets are as follows:
>
> | **Method**  | **Mean F1** |
> |--------------------------|:---------:|
> | GraphSMOTE (kNN)         | 0.681   |
> | GraphENS (kNN)           | 0.655   |
> | **GILA (ours)**          | **0.772** |
>
> Our method achieves substantially higher performance in these experiments. This demonstrates that the kNN graph is not an optimal structure for tabular data. GILA learns both the task-specific graph structure and the helper nodes, resulting in significantly better performance.
>
> **(8) Availability of Standard Deviations**
>
> Due to page limitations in the main paper, we were unable to include the full set of standard deviations directly in the main tables, and therefore placed them in the Appendix instead. As indicated in the caption of Table 2:
> “Corresponding standard deviations are reported in Appendix A.4.”
> Since this information may have been easy to overlook, we will highlight this reference more clearly in the revised version to avoid any confusion.
>
> **(9) Complexity Analysis with Helper Nodes**
>
> To address the reviewer’s concern regarding the computational and memory cost introduced by helper nodes, we provide a detailed complexity analysis below.
> Let **N** denote the number of original nodes, **M** the number of helper nodes, and **d** the feature dimension (shared by both original and helper nodes).
>
> After adding **M** helper nodes, the total number of nodes becomes **N + M**.
> The complexity of learning the adjacency matrix changes from **O(N² d)** to **O((N + M)² d)**, which expands to:
>
> (N + M)² d = (N² + 2NM + M²) d.
>
> In our setting, helper nodes are introduced only for the minority class to roughly balance the majority and minority classes.
> Since the minority class is never empty, the number of helper nodes **M** is strictly smaller than the original number of nodes, i.e., **M < N**.
> This implies:
>
> N < N + M < 2N,
>
> and thus the adjacency-learning complexity is upper bounded by:
>
> O((N + M)² d) < O((2N)² d) = O(4 N² d).
>
> That is, our method increases the cost by at most a constant factor (strictly less than 4× in this setting), while preserving the same asymptotic order **O(N² d)** as the original IDGL.
>
> The helper nodes are learnable, but updating their embeddings only costs **O(M d)**, which is insignificant relative to the adjacency-learning cost **O((N + M)² d)**.
>
> Therefore, the overall computational complexity remains of the same order as the original IDGL, with a moderate constant-factor overhead, while providing improved performance on imbalanced classes.
>
> **(10) Classifier Choice**
>
> GILA is not restricted to GCN; the method is compatible with various GNN architectures such as GAT, GraphSAGE, and GIN. However, the core contribution of this work lies in graph structure learning and the helper-node mechanism, not in the choice of classifier. While the optimal classifier may vary by dataset, we do not expect the choice of GNN architecture to fundamentally alter the conclusions. For fairness and controlled comparison, we used GCN, following the original IDGL setting.
>
> **Continued in the next comment.**

---

> ### Author Response · Authors · 2025-11-20
>
> **(11) Interpretation of helper node representations**
>
> In imbalanced classification, the classifier is often biased toward the majority class. To mitigate this effect, we assign the minority label to the helper nodes so that they provide supervision signals favoring the minority side. However, the helper nodes are **not** encouraged to model or approximate the minority data distribution. Their role is not to act as additional minority samples, but rather to serve as **embedding-level modifiers** that make the representations of the two classes more separable during graph structure learning.
>
> As shown in Figure 5, the helper nodes form connections with minority nodes while also maintaining a notable number of edges to majority nodes.
>
> Because they interact with both classes during graph structure learning, they influence the representations of both groups, contributing to increased separability between them.
>
> For this reason, their embeddings do not fully overlap with the minority cluster but instead appear in the intermediate region between the two classes.
>
> **(12) Clarification on multi-class settings**
>
> We would like to clarify that **our work focuses exclusively on binary classification**, following the standard setting used in prior studies such as CDBH and SSHR. All 44 tabular datasets included in our benchmark are *binary* and have small sample sizes, which aligns with the scope and motivation of GILA.
> Small-scale multi-class tabular datasets (with fewer than ~1000 samples) are extremely rare, and therefore multi-class scenarios were outside the scope of this study. Extending GILA to multi-class settings would require additional design choices—such as defining per-class auxiliary nodes or grouping strategies—which we consider valuable directions for future work.
> To avoid any ambiguity, we will explicitly state in the revised version that our method and experimental setup target binary imbalanced classification.
>
> **(13) Additional analysis**
>
> Thank you for raising this point. While we agree that theoretical analysis could further strengthen a method, we would like to highlight that our paper already provides extensive empirical evidence supporting the effectiveness and robustness of GILA. In addition to evaluations on 44 small-scale tabular datasets, we offer several complementary analyses—such as comparisons between IDGL and GILA, embedding and graph-structure visualizations, class-overlap robustness evaluation, and comparisons with class weighting and oversampling baselines—all consistently demonstrating the effectiveness of helper nodes.
>
> To further substantiate the mechanism of helper nodes, we provide an additional analysis. We train a linear SVM on the final node embeddings and measure the resulting decision margin. The results below show the average performance across all 44 datasets:
>
> | **Setting**          | **Accuracy** | **Margin_mean** |
> |:--------------------------------|:------------:|:----------------:|
> | w/o helper node      | 0.858        | 1.995            |
> | w helper node     | 0.902        | 2.604            |
>
> As shown, adding helper nodes significantly increases the inter-class margin. This supports our hypothesis that helper nodes influence the message-passing aggregation process such that the embeddings of different classes become more separable.
>
> Thank you again for your valuable feedback.
> We hope our responses address your concerns, and we remain open to further clarification during the discussion phase.

---

### Official Review · Reviewer_vBxW · 2025-11-02

**Soundness:** 2
**Presentation:** 3
**Contribution:** 2
**Rating:** 6
**Confidence:** 3

**Summary:**

This paper introduces GILA, a novel Graph Structure Learning (GSL) framework designed to handle class-
imbalanced tabular datasets — a scenario common in many real-world applications but largely
overlooked in current GSL research.
The method first constructs a graph from tabular data using IDGL as the structure learner and then
introduces helper nodes, pseudo-nodes that represent minority-class samples. These helper nodes are
learnable through an updater mechanism and influence minority nodes during message propagation to
enhance their separability from majority nodes. The authors conducted extensive experiments over many
datasets, demonstrating substantial gains over existing baselines (MLP, GCNkNN, SUBLIME, HES, IDGL).
Visualizations and correlation analyses further show that GILA is robust to class overlap and promotes
clear minority–majority separation in embedding space.

**Strengths:**

Introducing helper nodes as trainable pseudo-samples to mitigate imbalance in GSL is conceptually
elegant and new. It moves beyond naive reweighting or oversampling by coupling the mechanism with
graph structure learning. The experiments are comprehensive. The authors not only compares against
many baselines but also investigates robustness to class overlap, the effect of helper node quantity, and
comparisons with traditional imbalance techniques (class weighting, SMOTE). The t-SNE and graph
topology visualizations (Figures 5–6) convincingly show how helper nodes progressively encourage
minority–majority separation.

**Weaknesses:**

1. GILA largely inherits its graph learning mechanism from IDGL. The novelty, therefore, lies primarily in the
helper-node strategy rather than a new structure learner. This raises questions about generality — would
the same helper-node principle work equally well with other GSL methods (e.g., HES or SUBLIME)?
Besides, there is no exploration about the initialization method of the helper nodes. Whether the full zero
initialization is the optimal way?
1. Intuitively, the helper nodes are close to the minor-class nodes and they share the same ground-
truth label. Their representation after training should be also close. But this is different from what's
shown in figure 5. Could you give some interpretation?
2. About the effect of increasing helper nodes for the minority class nodes, the performance increasing
trend doesn't saturate as shown in figure 7. Would it continue to increase if adding more helper
nodes to make minority class nodes more than the majority class nodes?
2. Compared to existing papers on Graph Neural Networks for Tabular Data Learning, it seems the author only introduce
the helper node to solve the imbalanced problem. The method is naive.

**Questions:**

see above.

---

> ### Author Response · Authors · 2025-11-20
>
> We sincerely appreciate your careful reading and your thoughtful, constructive comments.
> We address each point below.
>
> **(1) Generality across different GSL backbones**
>
> Thank you for your thoughtful comment. Your question regarding the generality of the helper-node mechanism is very reasonable and aligns with the considerations we had during the selection of our GSL backbone.
>
> In fact, we also found this question important, and therefore we tested the helper-node idea on HES and SUBLIME during our backbone selection process. However, in both methods, the helper nodes contributed little to no performance improvement.
>
> Unlike IDGL, which updates the graph structure and classifier together within the same epoch through a **co-training** strategy, SUBLIME adopts a **two-stage** strategy in which the graph structure is pretrained first and the classifier is trained afterward. Therefore, SUBLIME appears unsuitable for training helper nodes, which must take both structure learning and classification into account simultaneously.
> HES learns the graph by increasing the homophily ratio. In this case, since the helper node has the same label as the minority class, many of its connections to the majority class will be pruned. However, in GILA, the helper node appears to connect to both minority and majority classes and contributes to increasing the separability between their embeddings. When strong homophily is enforced in HES, it becomes difficult for the helper node to maintain its connections to the majority class, preventing it from performing the intended role of enhancing inter-class separability.
> For these reasons, rather than proposing a universal helper-node mechanism applicable to any GSL method, we chose to develop our model based on the IDGL backbone, where helper nodes can be effectively trained.
>
> **(2) Initialization strategies for the helper nodes.**
>
> We explored an alternative initialization strategy for the helper nodes beyond the zero-vector initialization. Specifically, we initialized helper nodes using SMOTE-generated synthetic minority samples so that they would start with feature values similar to actual minority instances.
>
> However, as shown in the table below (average F1-score over 44 datasets), this approach generally **reduced performance** across most datasets. As discussed in Response 1, initializing helper nodes with minority-like features causes them to become **overly biased toward the minority class**, which limits their ability to interact with both classes and weakens their intended role of improving embedding separability.
>
> Although a few datasets showed slight improvements with SMOTE initialization, **zero-vector initialization** was consistently more stable and achieved superior overall performance. Therefore, we adopt zero initialization as our default design choice.
>
> We will include the corresponding comparison results in the Appendix.
>
> | Initialization | Mean F1-score | Count |
> |----------------|:---------:|:--------:|
> | SMOTE init.     | 0.623   | 10     |
> | Zero init.      | **0.772**   | **34**     |
>
> **(3) Interpretation of helper node representations**
>
> In imbalanced classification, the classifier is often biased toward the majority class. To mitigate this effect, we assign the minority label to the helper nodes so that they provide supervision signals favoring the minority side. However, the helper nodes are **not** encouraged to model or approximate the minority data distribution. Their role is not to act as additional minority samples, but rather to serve as **embedding-level modifiers** that make the representations of the two classes more separable during graph structure learning.
>
> As shown in Figure 5, the helper nodes form connections with minority nodes while also maintaining a notable number of edges to majority nodes. Because they interact with both classes during graph structure learning, they influence the representations of both groups, contributing to increased separability between them. For this reason, their embeddings do not fully overlap with the minority cluster but instead appear in the intermediate region between the two classes.
>
> **Continued in the next comment.**

---

> ### Author Response · Authors · 2025-11-20
>
> **(4) Effect of Increasing Helper Nodes**
>
> Thank you for the insightful question.
>
> We performed an additional experiment where we added more helper nodes than needed for class balancing.
>
> The table below reports the average F1 score across the 44 datasets.
>
> We found that performance actually **decreased** when too many helper nodes were added. Our interpretation is that when helper nodes become overly numerous, the **artificial signals** they introduce grow too strong and begin to dilute the meaningful signals coming from the original data samples.
>
> | Ratio| Mean F1 Score |
> |------------------|:-------------:|
> | 1.0              |     0.772     |
> | 1.1              |     0.687     |
> | 1.2              |     0.668     |
>
> **(5) Clarifying the contribution of helper nodes**
>
> GILA’s helper nodes are not synthetic minority samples, nor are they designed to approximate the minority feature distribution. Instead, they function at the embedding level by adjusting the representations of both classes and promoting clearer separation. To the best of our knowledge, this mechanism—where auxiliary nodes act not as data samples but as structural and embedding-level modifiers—has not been explored in prior GSL or imbalanced-learning literature, and thus represents a novel aspect of our approach.
>
> As discussed in Response 1, simply adding extra nodes is not sufficient; helper nodes require specific conditions to be effective in tabular GSL. Identifying these conditions, integrating them into a unified model, and validating them empirically also constitute an important contribution of our work.
>
> We appreciate your insightful feedback.
> We hope that our responses have clarified the key points, and we are happy to provide any additional details if needed.

---

### Author Response · Authors · 2025-12-01

We thank the reviewers for their time and constructive feedback. Below, we provide a brief summary addressing the major points.

We believe that some of the reviewers' concerns arise from a conceptual mismatch between our problem setting and the standard imbalanced graph learning setup. In (1), we clarify this distinction, and in (2), we provide additional empirical evidence to support our methodology.

**(1) Difference in problem setting and the role of additional nodes**

Our problem setting is different from imbalanced graph learning. We start from tabular data where **no graph is given**, and unlike methods that add synthetic minority samples, our helper nodes are **not data points** but structural components that promote class separation.
When tabular data is converted into a graph using the simplest approach—constructing a kNN graph—our method achieves substantially superior performance compared to existing imbalanced graph methods. This indicates that kNN is not an optimal graph structure, and that jointly learning the task-optimal graph and helper nodes, as in our approach, is far more effective for tabular data.

| Method  | Mean F1 |
|--------------------------|---------|
| GraphSMOTE (kNN)         | 0.681   |
| GraphENS (kNN)           | 0.655   |
| **GILA (ours)**          | **0.772** |

**(2) Additional Empirical Validation**

Using a linear SVM to evaluate the final node embeddings, we observed that adding helper nodes substantially improves both performance and inter-class margins. These results demonstrate that helper nodes meaningfully enhance class separability through the message-passing process.

| Setting             | Accuracy | Mean Margin |
|---------------------|----------|-------------|
| w/o helper node     | 0.858    | 1.995       |
| **w helper node**    |**0.902**    | **2.604**       |

---

### Author Response · Authors · 2025-12-02

The manuscript has been revised and re-uploaded. The main changes are as follows:

- **Updated Figure 2.**

    The edges in the graph illustration were changed from solid to dashed lines to clearly indicate that the graph structure is learned rather than fixed.


- **Added Section 4.6: Quantifying the Effect of Helper Nodes on Class Separation.**

    We included a new subsection providing quantitative evidence of improved class separability using SVM decision margins.


- **Added Appendix A.5: Comparison with Existing Imbalanced GNN Methods.**

    We added additional experiments comparing GILA with imbalanced graph learning models on kNN-constructed graphs to further contextualize the problem setting.

---

### Meta-Review · Area_Chair_Z5W6 · 2025-12-09

**Summary:**

Thanks to the reviewers for their valuable comments from many different perspectives. Overall, I think their main problems at present lie in:



- Lack of generality of the method.

- Lack of contribution and novelty.

- Parts of the description are unclear.

- The experimental section is inadequate.



In addition, some reviewers mentioned issues such as missing baselines.

**Reviewer Concerns:**

I would like to thank the authors for their responses, and I think some of the reviewers' questions will be addressed, such as some detailed explanations.

However, after considering the comments of the reviewers together with the responses of the authors, I believe that the responses of the authors regarding generality and novelty do not convince the reviewers. In addition, the doubts raised by the reviewers regarding the experimental part have not been addressed head-on, so I believe that the paper may not meet the ICLR acceptance criteria for the time being.



Overall, I think the paper fails to convince most reviewers and is below the acceptance threshold.

**Reviewer Scores:**

For Reviewer vBxW, clarification about generality and contribution may not work. I think this reviewer will maintain the score (**Rating:** 6).



For Reviewer Sjt8, doubts about novelty do not seem to work. I think this reviewer will maintain the score (**Rating:** 2).



For Reviewer e9XP, doubts about novelty do not seem to work. I think this reviewer will maintain the score (**Rating:** 2).

---

### Decision · Program_Chairs · 2026-01-26

Reject